# Hot Deformation Behavior and Microstructure Evolution of 6063 Aluminum Alloy Modified by Rare Earth Y and Al-Ti-B Master Alloy

**DOI:** 10.3390/ma13194244

**Published:** 2020-09-23

**Authors:** Wanwu Ding, Xiaoxiong Liu, Xiaoyan Zhao, Taili Chen, Haixia Zhang, Yan Cheng, Huaixin Shi

**Affiliations:** 1School of Materials Science and Engineering, Lanzhou University of Technology, Lanzhou 730050, China; lxxlbj@163.com (X.L.); zxyxmy@163.com (X.Z.); chen_taili@163.com (T.C.); zhanghx416@163.com (H.Z.); chengyanLUT@163.com (Y.C.); SHX15214026609@163.com (H.S.); 2State Key Laboratory of Advanced Processing and Recycling of Non-ferrous Metals, Lanzhou University of Technology, Lanzhou 730050, China

**Keywords:** 6063 aluminum alloy, hot deformation behavior, constitutive equation, processing map, microstructure evolution

## Abstract

The hot deformation behaviors of the new 6063 aluminum alloy modified by rare earth Y and Al-Ti-B master alloy were studied through isothermal hot compression experiments on the Gleeble-3800 thermal simulator. By characterizing the flow curves, constitutive models, hot processing maps, and microstructures, we can see from the true stress–true strain curves that the flow stress decreases with the increase of deformation temperature and the decrease of strain rate. Through the calculation of the constitutive equation, we derived that the activation energy of the new composite modified 6063 aluminum alloy is 224.570 KJ/mol. we roughly obtained its excellent hot processing range of temperatures between 470–540 °C and the strain rates of 0.01–0.1 s^−1^. The verification of the deformed microstructure shows that with the decrease of lnZ, the grain boundary changes from a low-angle one to a high-angle one and the dynamic recrystallization is dominated by geometric dynamic recrystallization and continuous dynamic recrystallization. Analysis of typical samples at 480 °C/0.01 s^−1^ shows that the addition of rare earth Y mainly helps form Al_3_Y_5_ and AlFeSiY phases, thus making the alloy have the performance of high-temperature recrystallization, which is beneficial to the hot workability of the alloy.

## 1. Introduction

As one of the most familiar Al-Mg-Si alloys, 6063 aluminum alloy not only has the basic properties of common aluminum alloys, it has also been widely used in building doors and windows, automotive, aerospace, and other fields because of its excellent hot workability and corrosion resistance. It is one of the most widely used deformed aluminum alloy materials [1,2,3]. In recent years, due to the deficiencies of the original 6063 aluminum alloy in extrusion processability and corrosion resistance, many studies on composition optimization have been carried out one after another. However, studies have shown that rare earth has extremely active chemical properties and adsorption, and rare earth and Al-Ti-B refiners are of great assistance to the grain refinement and the improvement of mechanical properties of aluminum alloys [4,5]. Wang et al. [6] studied the effect of mixed rare earth La and Ce on the structural and mechanical properties of Al-Mg-Si alloys, and the results show that the addition of mixed rare earth La and Ce has a positive effect on the grain refinement of the alloy, and that when the addition amount reaches 0.5%, the tensile strength and elongation of the alloy are significantly improved. Ma et al. [7] studied the microstructure and refining properties of Al-Ti-B-Er refiner. Compared with Al-Ti-B refiner, the addition of Er can help change the morphology of TiAl_3_ and TiB_2_ phases, which has an excellent effect on grain refinement of the alloy and can also help improve the mechanical properties of the alloy. Therefore, the author tried to add Al-Ti-B refiner and rare earth Y to 6063 aluminum alloy to achieve the goal of improving the properties of 6063 alloy.

As far as 6063 alloy is concerned, hot deformation is its main manufacturing technology, especially for extruding products. The product quality is not only related to their own properties, but also to their hot processing technologies. At present, there have been many studies reporting the use of thermal simulation methods to study the hot deformation characteristics of various alloys. Gan et al. [8] studied the hot deformation behaviors of 6063 aluminum alloy under high temperature conditions and established the high temperature flow stress model of the material. Yu Zhuhua and others [9] studied the hot compression deformation behaviors and structural evolution of 6063 aluminum alloy after homogenization, and they drew a hot processing map, analyzed the microstructures of the safe area and the unstable area, and thus obtained suitable process parameters of hot deformation. Babaniaris et al. [10] studied the effect of Al_3_(Sc, Zr) on the hot deformation behavior of 6-series alloys. It was found that the dispersed phase pinned dislocation movement and hindered the dynamic recrystallization. At the same time, the flow stress of Al-Mg-Si-Sc-Zr alloy was better than that of other 6-series alloys. Shi et al. [11] explored the dynamic softening mechanism of Al-Zn-Mg-Cu alloy during hot compression. The results show that the only softening mechanism is dynamic recovery at low temperature, while discontinuous dynamic recrystallization and continuous dynamic recrystallization exist at higher temperature. From the above reports, the studies on the hot deformation behaviors of 6063 aluminum alloy under different processes have been relatively mature, but the suitable hot working technology of the new 6063 aluminum alloy modified by rare earth Y and Al-Ti-B master alloy and the function mechanism of the addition of rare earth Y during the process of hot deformation of the alloy are still unclear.

In order to provide theoretical support for the practical application and production of the optimized new composite modified 6063 aluminum alloy, the extrusion processability of the modified alloy was explored, and the optimum hot working process parameters during hot deformation were further studied. In this paper, we carried out an isothermal compression experiment on the new 6063 aluminum alloy modified by rare earth Y and Al-Ti-B master alloy, studied the effects of temperature and strain rate on the hot deformation process. We constructed the Arrhenius constitutive model in consideration of strain compensation and drew the hot processing map based on the dynamic material model. We preliminarily established a suitable hot working range, then observed the deformed structure and analyzed and verified the obtained suitable hot working parameters, providing important guidelines for the practical application of the new composite modified 6063 aluminum alloy.

## 2. Materials and Methods

The materials used in the experiment were self-made new composite modified 6063 aluminum alloy, which was mainly synthesized from 6063 aluminum alloy, Al-Ti-B master alloy and rare earth Al-20Y. The chemical composition of the alloy studied is given in Table 1. The first step of the alloy preparation process was to melt the 6063 alloy in a crucible, and then add 0.5% refining agent for slag removal. When the temperature reached 730 °C, 0.2% Al-Ti-B and 0.5% Al-20Y were added simultaneously and the mixture was mechanically stirred for 1 min to make it mix uniformly, and then it was kept warm for 15 min, and finally casted into Φ 14 × 120 bar samples.

Due to some unfavorable factors caused by casting and the fact that hot working deformation requires the alloy to be homogenized, the cast sample was homogenized at a temperature of 568 °C for 4 h and air-cooled to room temperature to improve the deformability of the alloy. Subsequently, the sample was processed into a cylindrical shape of Φ 10 × 15 mm for thermal compression experiment. Figure 1a is the microstructure of the as-cast structure after corrosion by Keller reagent (2.5 mL HNO_3_ + 1.5 mL HCL + 1 mL HF + 95 mL H_2_O). It is obvious that there are some coarse dendrites and dendritic segregation, while Figure 1b is the microstructure of the homogenized sample that has been corroded by 40% hydrofluoric acid. It can be seen that the structure has no obvious grain boundary segregation and it is basically composed of uniform equiaxed crystals.

The cylindrical samples were subjected to thermal compression experiments on the Gleeble-3800 thermal simulator at a temperature range of 300–540 °C with intervals of 60 °C, strain rates of 0.01 s^−1^, 0.1 s^−1^, 1 s^−1^, and 10 s^−1^, and a true strain of 0.8 (with a deformation degree of about 55%). In order to make the data collected in the experiment be of unidirectional pressure and to avoid the influence of the deformation process due to excessive friction between the sample and the indenter, a method of lubricating graphite sheets at both ends of the sample was adopted. The sample was heated to the experimental temperature at a heating rate of 5 °C per second, and then it was preserved for 5 min to eliminate the temperature gradient before the experiment was started. The experimental process would automatically record the true stress and true strain values during the deformation process, and the deformed sample was immediately water quenched to retain the deformed structure. The deformed sample was cut in half along the central axis. One half was mounted, ground, and polished before being corroded with 40% hydrofluoric acid aqueous solution and its microstructure was observed with an optical microscope (OM) to study the evolutionary law of the structure during deformation. The other half was cut into thin slices with a thickness of 0.3 mm, ground to 70 μm and punched into small round slices of Φ 3 mm with a punching machine. The thin slices were put into GatanPIPS-695 ion thinning instrument, and ion thinned in a solution of 10% perchloric acid (HClO_4_) and 90% absolute ethanol (C_2_H_6_O) until the thin area appeared. Then analyzed by a transmission electron microscope (TEM; JEM-2010) for its internal mechanism of the deformation process.

## 3. Results and Discussions

### 3.1. Flow Stress Characteristics

The true stress–true strain curves (as shown in Figure 2) of the composite modified 6063 aluminum alloy at a deformation temperature range of 300–540 °C and a strain rate range of 0.01–10 s^−1^ can be obtained from the experimental data of isothermal compression. It can be seen from the figure that the flow stress decreases with the increase of the deformation temperature at the same strain rate; the flow stress increases with the increase of the strain rate at the same temperature [12,13,14]. The plastic deformation process of the aluminum alloy is mainly composed of work hardening and dynamic softening. In the initial stage of thermal deformation, the flow stress increases rapidly. This is due to the work hardening caused by the sharp increase in the dislocation density, which makes the dislocations entangle and interlace each other, hindering their movement. As the amount of strain increases, the deformation energy accumulated during the deformation process will reach the critical value of the material and become the driving force for dynamic recovery and dynamic recrystallization, which will cause the material to soften. It is shown by the curve as a smooth state. When the deformation continues and the softening rate is greater than the hardening rate, the curve will have a downward trend. In the actual plastic deformation process, the flow stress can not only reflect the microstructural evolution law and deformation mechanism, it can also measure the excellence in its processing under specific technology [15,16,17]. At higher strain rates of 1 s^−1^ and 10 s^−1^, there is no obvious softening phenomenon of the curve. On the one hand, it is because the strain rate is so high that it is too late for the dislocation entanglement and others produced to recover, so the hardening is still dominant; on the other hand, it is because the new precipitation phase generated by the combination between the rare earth Y and the original phase is distributed in the grain boundaries and within the grains, making it difficult for dislocations to climb and cross slip. For low strain rates, the mutual movement between dislocations will have sufficient time to recover, as shown in Figure 2a at 420 °C/0.01 s^−1^, after reaching the peak stress, the flow stress remains stable, reflecting the phenomenon of dynamic recrystallization, and the addition of rare earth Y can help provide nucleation sites for dynamic recrystallization, which can promote the occurrence of dynamic recrystallization, so it is conducive to structural homogenization, grain refinement, etc., and can help improve the strength and toughness of the product.

### 3.2. Construction of Constitutive Equation

The constitutive model is considered to be an important mathematical method to describe the flow behaviors of alloys during their plastic deformation. It can express the internal relationship between flow stress (*σ*), deformation temperature (*T*), and strain rate ($\dot{\varepsilon }$) [18,19,20]. It is usually expressed by the hyperbolic sine function Arrhenius equation proposed by Sellars and Tegart, which includes the deformation temperature and the strain activation energy (*Q*), and at the same time, the temperature compensation strain rate factor *Z* (Zener–Hollomon) is also taken into consideration [16,21,22,23]
(1)$$Ζ=\overset{•}{\varepsilon }\exp\left[Q/\left(RT\right)\right]$$
(2)$$\overset{•}{\varepsilon }=A_{1}\sigma^{n_{1}}\exp\left[-Q/\left(RT\right)\right]\alpha \sigma <1.2$$
(3)$$\overset{•}{\varepsilon }=A_{2}\exp\left(\beta \sigma \right)\exp\left[-Q/\left(RT\right)\right]\alpha \sigma >1.2$$
(4)$$\overset{}{\varepsilon }=A{\left[sinh\left(\alpha \sigma \right)\right]}^{n}\exp\left[-Q/\left(RT\right)\right]\;\mathrm{for}\;\mathrm{all}\;\sigma \text{’}s$$

Among them, *A*, *A*_1_, *A*_2_, *n*, *n*_1_, and *β* are all constants, and the stress level parameter *α* = *β*/*n*, and *R* is the gas constant (8.314 J/mol/K). Take the logarithm of both sides of Equations (2) and (3) and we will get
(5)$$\ln\overset{}{\varepsilon }=\ln A_{1}{+n}_{1}\ln\sigma -\left(\frac{Q}{RT}\right)$$
(6)$$\ln\overset{}{\varepsilon }=\ln A_{2}+\beta \sigma -\left(\frac{Q}{RT}\right)$$

Under different conditions, according to the peak flow stress, the method of linear regression can be used to fit the relationship between *ln*$\dot{\varepsilon }$-*ln**σ* and *ln*$\dot{\varepsilon }$-*σ*, and the values of *n*_1_ and *β* can be obtained. The fitting results are shown in Figure 3a,b. The mathematical conversion of Equation (4) is
(7)$$Q=R{\frac{\partial \ln\overset{}{\varepsilon }}{\partial \ln\left[\sin h\left(\alpha \sigma \right)\right]}|}_{T}{\frac{\partial \ln\left[\sin h\left(\alpha \sigma \right)\right]}{\partial \left(\frac{1}{T}\right)}|}_{\overset{}{\varepsilon }}=RnK$$

It can be seen from Equation (7) that when the strain rate and deformation temperature are constant, ln$\dot{\varepsilon }$ has a linear relationship with *ln*[*sinh*(*ασ*)] and 1/T. Therefore, the two can be linearly regressed respectively by the least square method. The slope is the value of *n* and *K*, and the fitting results are shown in Figure 3c,d. From Equations (1) and (4), we can see that *lnZ* and *ln*[*sinh*(*α**σ*)] also have a linear relationship, and the intercept after fitting is the value of *lnZ*, as shown in Figure 4. Summing up the calculation results, we get the constitutive equation of composite modified 6063 aluminum alloy
(8)$$\overset{•}{\varepsilon }=3.45\times 10^{16}{\left[\sin h\left(0.0234\sigma \right)\right]}^{7.231}\exp\left[\frac{-224.570}{\left(RT\right)}\right]$$

The establishment of the constitutive model is helpful to predict the flow stress of the material during the thermal deformation process and can provide a theoretical basis for the subsequent production and processing. The activation energy *Q* is an important physical parameter to measure the thermal deformation process of aluminum alloy, which is the common effect between the deformation temperature, the strain rate, and the flow stress [24,25,26]. The calculated diffusion activation energy *Q* of the composite modified 6063 aluminum alloy is 224.570 KJ/mol, which is greater than 165 KJ/mol, the self-diffusion activation energy of pure aluminum and greater than 175 KJ/mol, the deformation activation energy of 6063 aluminum alloy. Therefore, the movement mechanism during its deformation is mainly the cross slip and climbing of dislocations, and the addition of rare earth Y will form pinning dislocations in fine dispersed phase, so dislocations need more storage energy during movement, resulting in high activation energy *Q* of composite modified 6063 aluminum alloy. The high activation energy indicates that the main softening mechanism in the thermal deformation process is dynamic recrystallization, the occurrence of which can not only cause the material to produce steady-state flow, which can be seen from the true stress–true strain curves, but it also helps refine the grains and improve the overall performance of the material [8,27,28].

### 3.3. Processing Map Analysis

The processing map was first proposed by Prasad and Sasidhara based on the dynamic material model to describe the energy change caused by the structural evolution of the material during the deformation process, and it is usually used to predict the excellent hot processing parameters of the material [29,30,31]. According to Prasad’s theory, in the hot processing process, the workpiece is regarded as a nonlinear power dissipator. The energy absorbed by the workpiece in the plastic deformation process is *P*, and the power that the environment input the deformed body is consumed in two aspects: (1) The energy *G* (power dissipation) consumed by plastic deformation of the material; (2) The energy *J* (power dissipation co-content) consumed by the microstructure evolution (such as dynamic recovery, dynamic recrystallization, and phase transition) during material deformation, and the relationship between them is as follows [25,29,32]
(9)$$P=\sigma \dot{\varepsilon }=J+G=\int_{0}^{\sigma }\dot{\varepsilon }d\sigma +\int_{0}^{\dot{\varepsilon }}\sigma d\dot{\varepsilon }$$

The proportion of the two energies is determined by the strain rate sensitivity factor *m* of the material under a certain stress.
(10)$$\frac{\partial J}{\partial G}=\frac{\dot{\varepsilon }\partial \sigma }{\sigma \partial \dot{\varepsilon }}=\frac{\dot{\varepsilon }\sigma \partial \ln\sigma }{\sigma \dot{\varepsilon }\partial \ln\dot{\varepsilon }}=m$$

At a certain temperature and strain, the stress and the strain rate of a hot processed workpiece have the following dynamic relationship
(11)$$\sigma =K{\dot{\varepsilon }}^{m}$$
where *K* represents the flow stress when the strain rate is 1, and the power dissipation co-content *J* can be expressed as
(12)$$J=\sigma \dot{\varepsilon }-\int_{0}^{\dot{\varepsilon }}K{\dot{\varepsilon }}^{m}=\frac{m}{m+1}\sigma \dot{\varepsilon }$$

It is generally considered that the material is in an ideal linear dissipation state, at this time *m* = 1, *J_max_* = *σ*$\dot{\varepsilon }$/2. The dimensionless parameter value *η*, which reflects the power dissipation of the material, is called the power dissipation factor, which can measure the energy dissipated by the structural evolution of the material during the deformation process.
(13)$$\eta =\frac{J}{J_{max}\frac{2m}{m+1}}$$

The criterion for flow instability of materials during plastic deformation is usually derived from the theory of maximum entropy productivity [33]
(14)$$\xi \left(\dot{\varepsilon }\right)=\frac{\partial \ln\left[m∕\left(m+1\right)\right]}{\partial \ln\dot{\varepsilon }}+m<0$$

At a certain strain rate and temperature, the value of *m* can be obtained by polynomial fitting, and the value of *η* can be calculated. Since the flow stress expressed by the material is different under different true strain values, here we take the true strain *ε* = 0.7. Figure 5a is a 3D map of power dissipation based on the power dissipation factor *η*, and Figure 5b is a 3D map of flow instability drawn in consideration of flow instability. The hot processing map is formed by superimposing the power dissipation map and the flow instability map in the same coordinate system. Figure 6 is the two-dimensional processing map with a true strain of 0.7.

The value of the contour in Figure 5a is the power dissipation factor η, which is a ternary variable about deformation temperature, strain rate and strain, so it can reflect the dynamic recovery, dynamic recrystallization and other plastic deformation mechanisms occurred in different areas of the structure [25]. Generally speaking, a high value of *η* indicates that the alloy consumes a high proportion of energy during structural transformation and the alloy has excellent workability. However, if the value of η is too high, deformation instability may also exist. Therefore, in order to avoid the structural deformation defects caused by flow instability, such as adiabatic shear bands, voids, cracks, and others, instability criteria are introduced [27]. From Figure 5a, it can be seen intuitively that the alloy has a high-power dissipation area at 430–540 °C and 0.01–1 s^−1^. To verify whether this area is a safe processing one, we can observe the corresponding instability values of the contour lines in the area in Figure 5b. If they are all positive, it indicates that it is in the safe processing area. In order to further determine the excellent processing range of the alloy, we can analyze through the two-dimensional plan view of the power dissipation map and the flow instability map in Figure 6. The value in the figure represents the power dissipation factor, and the shadow part represents the area of flow instability. Integrating the conditions of power dissipation and flow instability, the approximate optimal processing range of the alloy is obtained at a temperature range of 470–540 °C and a strain rate range of 0.01–0.1 s^−1^. Further verification can be obtained through subsequent structural analysis.

### 3.4. Structural Evolution during Deformation

Exploring the changes in the microstructure of the alloy during its plastic deformation is essential to further determine the hot working parameters of the alloy. At the same time, many studies have shown that the temperature compensation strain rate factor *Z* is closely related to the microstructure during hot deformation. From Equation (1), it can be seen that the value of *Z* is related to the deformation temperature and the strain rate. Therefore, according to the changes in the value of *Z*, we chose some typical samples to analyze the microstructures during their deformation.

Figure 7 shows the microstructures of the alloy under different deformation conditions. Compared with the homogenized as-cast structure, when *lnZ* = 49.44 (300 °C/10 s^−1^), the thermally deformed grains are obviously flattened, elongated, and become fibrous along the vertical stress direction. This is because the deformation temperature is low and the strain rate is high. The crystal grains are too late to recover in a short time. The dislocation density increases rapidly. The interaction between the dislocations hinders their movement and produces work hardening, so it has a larger flow stress [24,25,26,27]. With the increase of the deformation temperature and the decrease of the strain rate, when *lnZ* = 42.67 (360 °C/1 s^−1^), the fibrous structure is obviously improved, and the dislocation storage energy reaches a certain degree and dynamic recovery occurs. At the same time, due to the effect of dynamic recovery, the dislocation density decreases, and the dislocations recombine and rearrange into the low-energy sub-grain boundaries, forming a sub-crystal structure surrounded by low-angle grain boundaries (LABs). When *lnZ* = 38.97 (420 °C/1 s^−1^), the grain boundaries are obviously serrated and the sub-grain boundaries become blurred [24,25,26,27]. When the temperature rises to 480 °C and the strain rate is 0.1 s^−1^ (*lnZ* = 33.56), the serration boundaries contact each other, the grains are pinched and broken into small new grains, and geometric dynamic recrystallization (GDRX) occurs. This nucleation mechanism is called grain boundary protrusion mechanism [34,35,36]. In addition, the increase in temperature and the decrease in strain rate can help increase the activation ability of atoms and the storage energy of dislocations, and the dislocations on some adjacent sub-grain boundaries form large sub-crystals through slipping and climbing. At the same time, through atomic diffusion and rotation of adjacent sub-crystals, the orientation of the two sub-crystals are aligned, realizing changes from low-angle grain boundaries (LABs) to high-angle grain boundaries (HABs), which is a typical continuous dynamic recrystallization (CDRX). This nucleation mechanism is called the subcrystal merging mechanism, and the dynamic softening mechanism also changes from dynamic recovery to dynamic recrystallization [34,35,36]. When *lnZ* continues to decrease, the increase in temperature is conducive to the movement of movable dislocations, and the absorption of dislocations by the grain boundaries becomes clear and straight to form more high-angle grain boundaries, which is conducive to the occurrence of dynamic recrystallization. Secondly, the strain rate decreases. The formed dynamic recrystallized grains have enough time to grow up, and finally most of the grains are equiaxed (*lnZ* = 31.26 (480 °C/0.01 s^−1^) and *lnZ* = 28.61 (540 °C/0.01 s^−1^)).

Figure 8 shows the transmission electron photos at different deformation temperatures and strain rates. The transmission electron photos are indispensable for in-depth analysis for the evolution law and mechanism of the structures during the alloy deformation process. As shown in Figure 8a, the deformation temperature is 300 °C and the strain rate is 10 s^−1^. A large number of dislocations and dislocation entanglements are distributed in the grain boundaries and within the grains, resulting in work hardening and blurring of the definition of grain boundaries. This is a typical condition of high *lnZ* values in all hot deformations. In the process of hot deformation, there is not only work hardening, but also dynamic softening [1,20,25]. When the alloy is at a low *lnZ* value, the increase of dislocation storage energy causes it to have sufficient mobility and the dislocation generation rate is less than its elimination rate, which promotes the sliding of the grain boundary and occurrence of dynamic recovery. The climbing of edge dislocations and the cross-slip of screw dislocations cause opposite sign dislocations to offset each other and same sign dislocations to repel one another, forming many dislocation cell walls. As the deformation progresses, the dislocation cell walls are polygonized, forming many sub-crystals surrounded by low-angle grain boundaries (arrow A in Figure 8b). When *lnZ* reduces, there is greater storage energy for the dislocations, and the slip and climb of dislocations are easier to proceed. The sub-crystals gradually grow, and the dislocation density will also be greatly reduced (arrow B in Figure 8c). At higher deformation temperatures, the thermal motion of atoms is enhanced, and the annihilation and rearrangement of dislocations proceed more fully. The absorption of dislocations by sub-grain boundaries promotes the transformation of low-angle grain boundaries to high-angle grain boundaries, and the reduction of strain rate can make it possible that there is enough time for the growth of subcrystals and the migration of dislocations. The subcrystals continue to merge with adjacent subcrystal boundaries to form large-sized subcrystals. During the process of subcrystal merging, rotation occurs, which gradually increases the angle of subcrystals to form high-angle grain boundaries (Figure 8d,e). The generation of high-angle grain boundaries (HABs) enables the occurrence of dynamic recrystallization [37,38,39].

Through the study of the evolution process of the structure, the internal grain and grain boundary changes of the alloy during the hot deformation process were known, and the approximate excellent processing range of the alloy was preliminarily verified. However, the analysis of the mode of action and the precipitated phase of the samples within the range is still lacking. Therefore, a typical sample with a temperature of 480 °C and a strain rate of 0.01 s^−1^ was selected for TEM analysis, as shown in Figure 9. Figure 9a is a high-angle annular dark-field image (STEM-HAADF) of particles inside a typical sample. The particles are similar to a packaged structure with a size of about 1 μm. The surface scanning shows that the structure is mainly composed of three parts, which is mostly distributed with Al, Fe, Si, and Y elements, and there is also a small amount of Mg element, and the addition of rare earth Y will make the Mg_2_Si particles in the strengthening phase smaller and make them distribute more uniformly.

In order to further analyze the structure qualitatively, electron diffraction spot calibration for selected area, high-resolution auxiliary phase, and point scanning energy spectrum analysis of the TEM bright field phase were carried out, as shown in Figure 10. Figure 10a shows the TEM bright field phase of the structure, which consists of three areas: A, B, and C. The electron diffraction for selected area was analyzed toward area A (Figure 10b). Through diffraction spot calibration the particle was derived as Al_3_Y_5_, a simple hexagonal structure, and its corresponding crystal belt axis is $\left[1\bar{1}0\right]$ its lattice constant a = 0.439 nm, and the lattice mismatch δ with α-Al is 8.6%. Figure 10c is the high-resolution auxiliary phase corresponding to area B and the fast Fourier transform (FFT) performed on it. At the same time, according to the point scanning energy spectrum corresponding to area B in Figure 10d, the phase was derived as AlFeSi. Similarly, the point scanning energy spectrum of area C corresponding to Figure 10e shows that it is an AlFeSiY compound.

It can be analyzed by the TEM surface scanning toward the distribution of each element and the corresponding point scanning toward energy spectrum that the addition of rare earth Y goes in two main directions. A part of Y forms Al_3_Y_5_ phase with Al, which has a partial coherent relationship with the α-Al matrix, so that it can be used as a heterogeneous nucleation core, causing the alloy to have the ability of DRX as well at higher temperatures; another part of Y will integrate with the original coarse AlFeSi and form small particles of AlFeSiY compound inside. Although it will disperse the larger AlFeSi, the formed fine particles will have a pinning effect on part of the grains and sub-grain boundaries and block the movement of the boundaries of recrystallized particles. This leads to an increase in the recrystallization temperature of the alloy, which has high temperature recrystallization performance, thus being beneficial to the hot workability of the alloy.

## 4. Conclusions

(1)The flow stress of the new 6063 aluminum alloy modified by rare earth Y and Al-Ti-B master alloy decreases with the increase of the deformation temperature and the decrease of the strain rate. After reaching the peak stress, the softening part of the curve tends to be relatively flat, which is manifested as a dynamic recrystallization mechanism.(2)The flow constitutive equation of the new 6063 aluminum alloy modified by rare earth Y and Al-Ti-B master alloy was established to be $\dot{\varepsilon }$ = 3.45 × 10^16^[sinh(0.0234σ)]^7.231^exp[−224.570/(RT)], which can provide prediction for the subsequent hot working process, and its self-diffusion activation energy *Q* is 224.570 KJ/mol, much higher than that of pure aluminum and 6063 aluminum alloy. The high *Q* value is mainly due to the addition of rare earths.(3)Based on the constructed two-dimensional and three-dimensional thermal processing map, considering the avoidance of instability defects, the approximately excellent processing range of the alloy can be obtained: temperature 470–540 °C, strain rate 0.01–0.1 s^−1^.(4)With the gradual decrease of lnZ, the alloy undergoes a process from initial work hardening to dynamic recovery and then dynamic recrystallization, which corresponds to the true stress–true strain curve. Dynamic recrystallization of the alloy is a common effect of geometric dynamic recrystallization (GDRX) and continuous dynamic recrystallization (CDRX).(5)Surface scanning of the high-angle annular dark field image (STEM-HAADF) shows that the particles are mainly composed of Al, Fe, Si, and Y elements at 480 °C/0.01 s^−1^. In addition, part of rare earth Y forms Al_3_Y_5_ with Al, which can be used as a heterogeneous nucleation core; the other part generates small particles of AlFeSiY inside AlFeSi, which will increase the recrystallization temperature of the alloy, making it have high temperature recrystallization performance.

## Figures and Tables

**Figure 1 materials-13-04244-f001:**
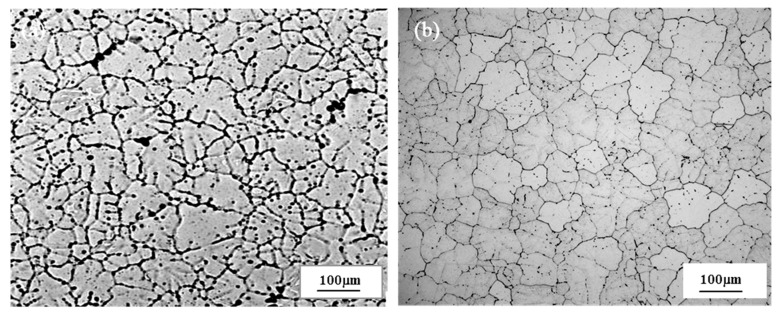
Microstructures of composite modified 6063 aluminum alloy: (**a**) as-cast; (**b**) homogenized.

**Figure 2 materials-13-04244-f002:**
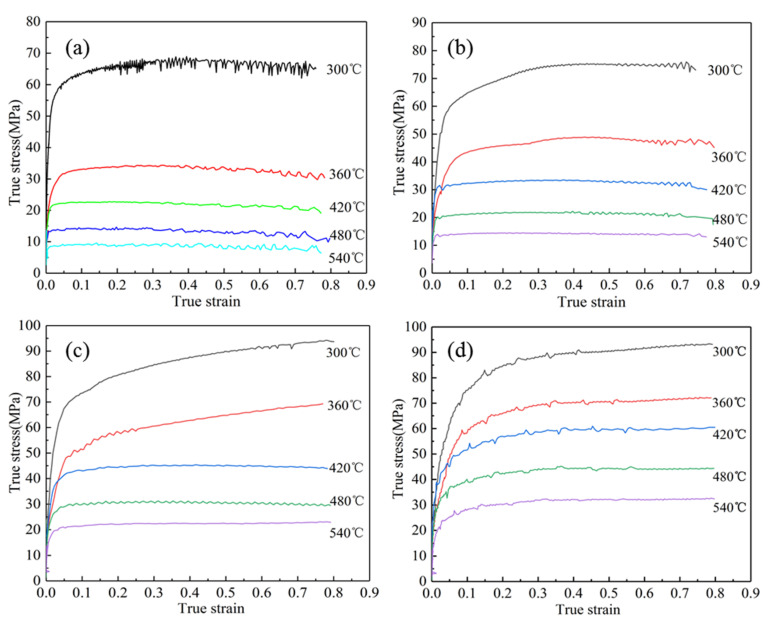
True stress–true strain curves of the composite modified 6063 aluminum alloy during hot compression: (**a**) $\dot{\varepsilon }$ = 0.01 s^−1^; (**b**) $\dot{\varepsilon }$ = 0.1 s^−1^; (**c**) $\dot{\varepsilon }$ = 1 s^−1^; (**d**) $\dot{\varepsilon }$ = 10 s^−1^.

**Figure 3 materials-13-04244-f003:**
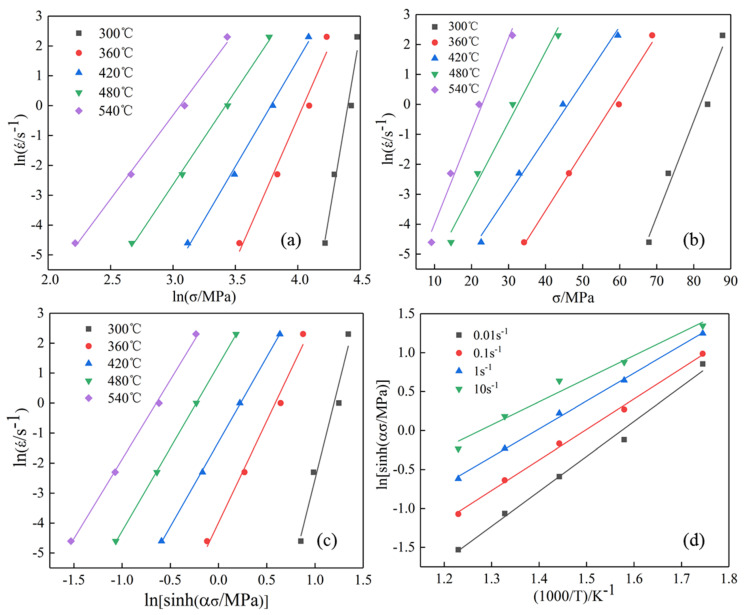
Linear fitting relationship of flow stress: (**a**) *ln*$\dot{\varepsilon }$-*lnσ*; (**b**) *ln*$\dot{\varepsilon }$-*σ*; (**c**) *ln*$\dot{\varepsilon }$-*ln*[*sinh*(*ασ*)]; (**d**) *ln*$\dot{\varepsilon }$ -1/*T*.

**Figure 4 materials-13-04244-f004:**
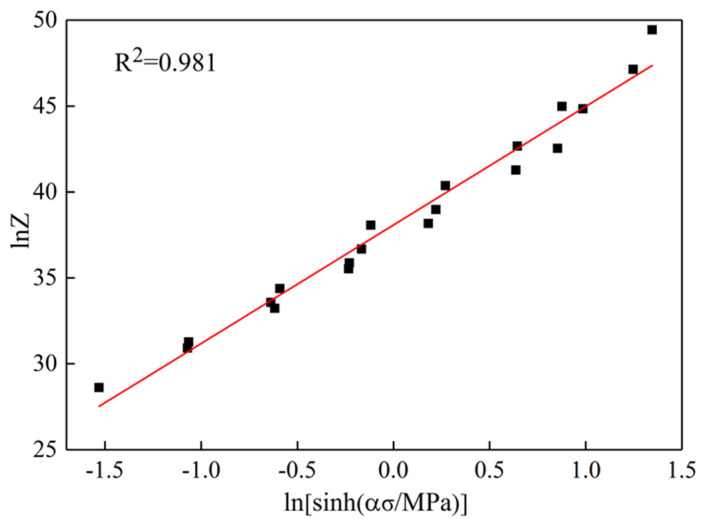
Linear fitting relationship between *ln*[*sinh*(*α**σ*)] and *lnZ*.

**Figure 5 materials-13-04244-f005:**
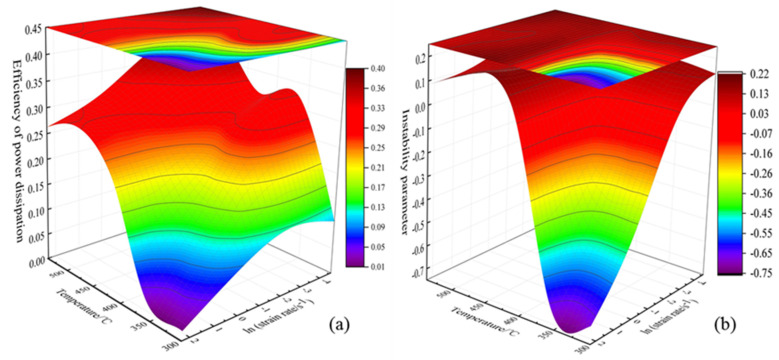
3D processing map drawn through power dissipation factor *η* with the true strain *ε* = 0.7. (**a**) power dissipation map; (**b**) flow instability map.

**Figure 6 materials-13-04244-f006:**
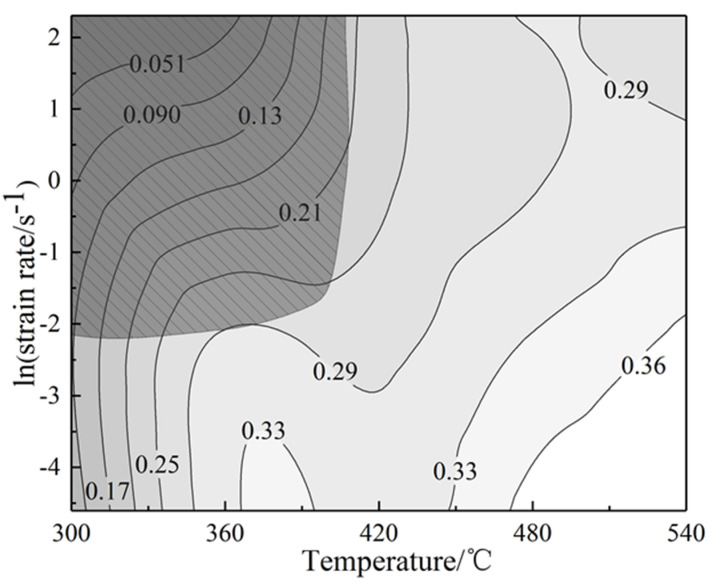
Two-dimensional hot working map of the composite modified 6063 aluminum alloy at a true strain of 0.7.

**Figure 7 materials-13-04244-f007:**
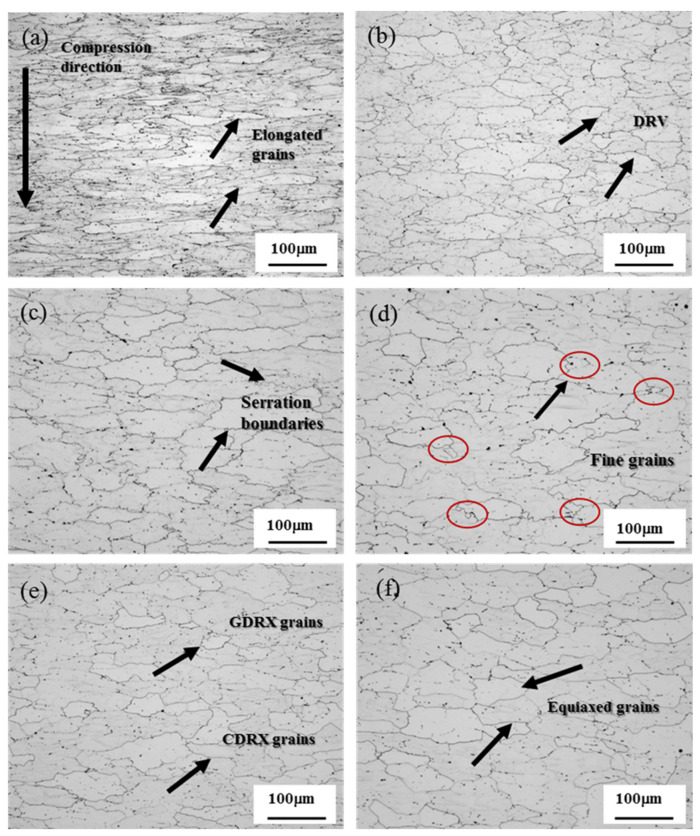
Microstructures of the alloy under different deformation conditions after hot compression: (**a**) 300 °C/10 s^−1^, *lnZ* = 49.44; (**b**) 360 °C/1 s^−1^, *lnZ* = 42.67; (**c**) 420 °C/1 s^−1^, *lnZ* = 38.97; (**d**) 480 °C/0.1 s^−1^, *lnZ* = 33.56; (**e**) 480 °C/0.01 s^−1^, *lnZ* = 31.26; (**f**) 540 °C/0.01 s^−1^, *lnZ* = 28.61.

**Figure 8 materials-13-04244-f008:**
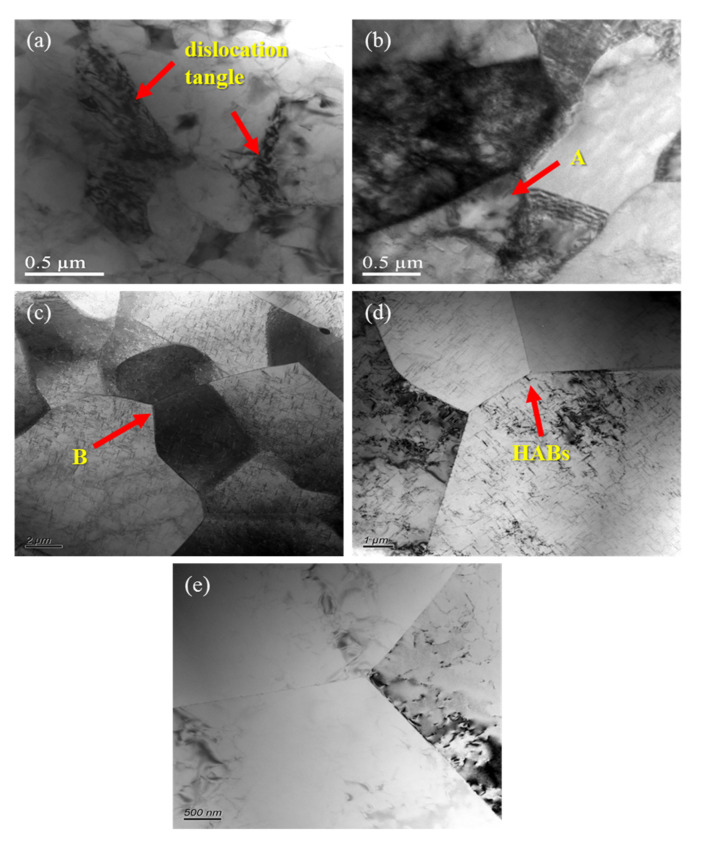
TEM deformed structures of the alloy at different temperatures and strain rates: (**a**) 300 °C/10 s^−1^, *lnZ* = 49.44; (**b**) 360 °C/1 s^−1^, *lnZ* = 42.67; (**c**) 420 °C/1 s^−1^, *lnZ* = 38.97; (**d**) 480 °C/0.01 s^−1^, *lnZ* = 31.26; (**e**) 540 °C/0.01 s^−1^, *lnZ* = 28.61.

**Figure 9 materials-13-04244-f009:**
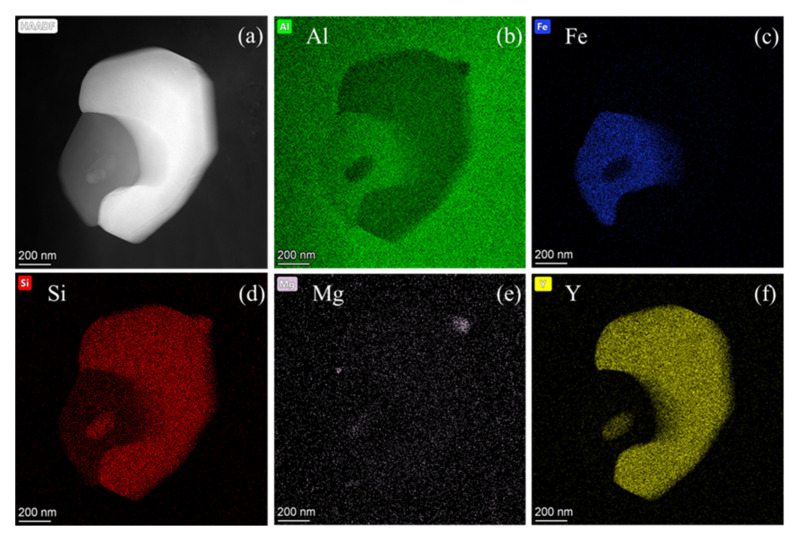
STEM-HAADF image and surface scanning distribution map of intragranular particles of a typical sample (480 °C/0.01 s^−1^) after hot compression: (**a**) STEM-HAADF image of intragranular particles; (**b**–**f**) surface scanning distribution of elements Al, Fe, Si, Mg, and Y.

**Figure 10 materials-13-04244-f010:**
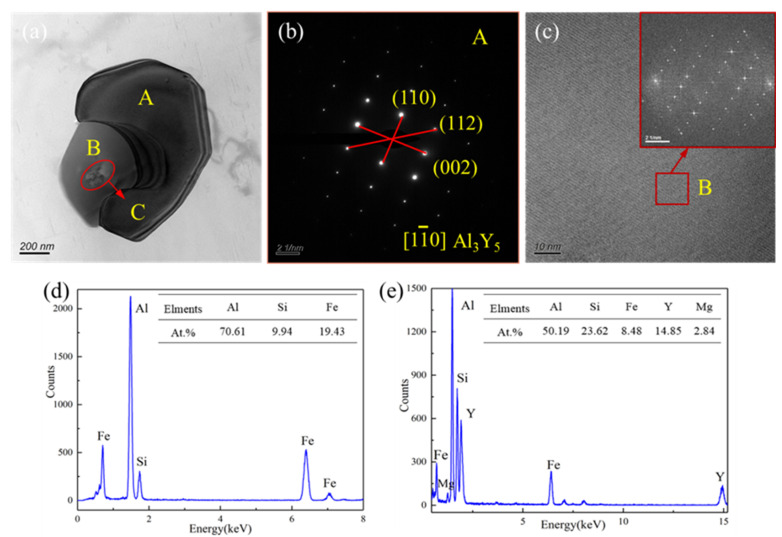
TEM bright field phase of the intragranular particles of the typical sample under the deformation condition of 480 °C/0.01 s^−1^ and the electron diffraction spots of the selected area, high-resolution auxiliary phase and energy spectrum of the corresponding area: (**a**) TEM bright field phase of particles; (**b**) selected area electron diffraction spots in area A; (**c**,**d**) high-resolution auxiliary phase and energy spectrum of area B; (**e**) corresponding energy spectrum of area C.

**Table 1 materials-13-04244-t001:** Chemical composition of the studied alloy (wt %)

Si	Mg	Fe	Cu	Ti	Y	Al
0.412	0.569	0.144	0.036	0.02	0.1	Bal.

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
