# Peer review of "Hot Deformation Behavior and Microstructure Evolution of 6063 Aluminum Alloy Modified by Rare Earth Y and Al-Ti-B Master Alloy"

_materials, 2020, doi:10.3390/ma13194244_

Round 1

Reviewer 1 Report

Several grammatical mistakes and poorly structured sentences. These should be addressed.

Why do rare earths improve alloy performance and refine grains? This should be introduced in the introduction. Work looking at hot deformation of Al alloys with additions of Zr, Sc and even older work on Mn additions by Qian et al, Babaniaris et al and Shi et al should be considered and cited. 

New Alloy compositions should be given. Not the composition of the additive alloys. This is extremely important. 

TEM experimental details are required.

While i do agree these are the driving mechanisms, It is very difficult to truly determine CDRX and GDRC from optical microscopy. EBSD is better in this regard.

The hot deformation performance of this alloy is notably worse than a traditional 6060 Al-Mg-Si alloy. More detail should be conducted on the Al3Y precipitates to confirm if they provide strength and information on their size and how they are dispersed in the microstructure is required. Looking at 1 particle is not enough. 

Author Response

Dear experts:

  Thank you very much! I feel very honored for that you have revised my manuscript and given good comments. I provide a cover letter to explain “point by point” the details of the revisions in the manuscript and please see the attachment.

Reviewer 2 Report

The manuscript presented by the authors fits perfectly with the scope of this Journal. However, prior to be accepted some small changes may be conducted in order to enhance te quality of it.

(1) Introduction; the contextualization of it may be improved. The reader must have clear why it is important to do this research.

(2) Materials and methods; it is necessary to specify the model of the machine; for exemple TEM.

(3) In figure 2, the y-axis should be true stress (MPa) instead of Mpa (please, take care with the units). furthermore, in the x-axis, it should be, true strain (-).

(4) The different equations may be re-written due to the quality of them is not appropiate, for example when the author describes epsilon (see equations 4 to 7

(5) The lines presented in Figure 3, should be dash (it is a fitting and in between two experimental points you do not have real value)

(6) The quality of the SEM micrographs (Figure 7) may be improved. Some times it is difficult to follow the description of the author.

(7) The conclusions of the manuscript are apropiate. However, they must be improved and go in depth with the main idea that the author would like to trasmit to the readers.

Author Response

(The authors gave the same response as above.)

Round 2

Reviewer 2 Report

The manuscript entitled Hot Deformation Behavior and Microstructure Evolution of 6063 Aluminum Alloy Modified by Rare Earth Y and Al-Ti-B Master Alloy is suitable to be published in Materials after conducting certain changes:

(1) The motivation of this manuscript may be included in the introduction. At the end, the author may try to answer the following question: Why it is important to conduct this research?

(2) The units of the stress is MPa no Mpa, it may be changed mainly in the figures present in the manuscript

(3) The author may summarize the conclusions. Some information present in it is supperfluous and it does not provide any additional information.

Author Response

Dear expert:

Thank you very much! I feel very honored for that you have revised my manuscript and given good comments. I provide a cover letter to explain “point by point” the details of the revisions in the manuscript and please see the attachment.
